# In Silico Analysis of the Longevity and Timeline of Individual Germinal Center Reactions in a Primary Immune Response

**DOI:** 10.3390/cells10071736

**Published:** 2021-07-09

**Authors:** Theinmozhi Arulraj, Sebastian C. Binder, Michael Meyer-Hermann

**Affiliations:** 1Department of Systems Immunology, Braunschweig Integrated Centre of Systems Biology, Helmholtz Centre for Infection Research, 38106 Braunschweig, Germany; theinmozhi.2910@gmail.com (T.A.); sb@theoretical-biology.de (S.C.B.); 2Institute for Biochemistry, Biotechnology and Bioinformatics, Technische Universität Braunschweig, 38106 Braunschweig, Germany

**Keywords:** germinal centers, agent-based modeling, primary immune response, germinal center kinetics, germinal center lifetime, germinal center contraction, vaccination, asynchronous germinal centers

## Abstract

Germinal centers (GCs) are transient structures in the secondary lymphoid organs, where B cells undergo affinity maturation to produce high affinity memory and plasma cells. The lifetime of GC responses is a critical factor limiting the extent of affinity maturation and efficiency of antibody responses. While the average lifetime of overall GC reactions in a lymphoid organ is determined experimentally, the lifetime of individual GCs has not been monitored due to technical difficulties in longitudinal analysis. In silico analysis of the contraction phase of GC responses towards primary immunization with sheep red blood cells suggested that if individual GCs had similar lifetimes, the data would be consistent only when new GCs were formed until a very late phase after immunization. Alternatively, there could be a large variation in the lifetime of individual GCs suggesting that both long and short-lived GCs might exist in the same lymphoid organ. Simulations predicted that such differences in the lifetime of GCs could arise due to variations in antigen availability and founder cell composition. These findings identify the potential factors limiting GC lifetime and contribute to an understanding of overall GC responses from the perspective of individual GCs in a primary immune response.

## 1. Introduction

Germinal centers (GCs) are transient anatomic structures in secondary lymphoid organs, responsible for affinity maturation of B cells, following an infection [1]. In the GCs, B cells undergo multiple rounds of mutation and selection resulting in the production of high affinity plasma and memory cells [2]. GCs are seeded by 10–100 B cell clones [3] and evolution of GC results in the formation of two compartments—a dark zone (DZ) and a light zone (LZ) [4,5]. In the DZ, GC B cells undergo proliferation and somatic hypermutation, a process where point mutations are randomly incorporated into the B cell receptor (BCR) genes leading to changes in affinities [6]. Selection of GC B cells occur in the LZ by interactions with follicular dendritic cells (FDCs) and T follicular helper (Tfh) cells that rescue GC B cells from apoptosis and trigger further rounds of cell divisions [7,8,9,10]. Alternatively, selected GC B cells differentiate into precursors of memory or plasma cells [11,12]. Multiple GCs are formed in a lymphoid organ in response to immunization, for instance, hundreds of GCs are formed in spleen after immunization with sheep red blood cells (SRBCs) [13] and 10–16 GCs are formed in single popliteal lymph nodes (LNs) after immunization with 4–Hydroxy-3-nitrophenylacetyl ovalbumin (NP-OVA) [14]. Memory and plasma cells arising from all the GCs collectively contribute to the humoral immune response.

GC reactions have a limited lifetime and are terminated by an unknown mechanism. GC reactions to certain antigens such as haptens are short-lived and have a lifetime of approximately 3 weeks [15,16]. On the other hand, viral infections can result in long-lasting or persistent GC reactions [16,17]. Memory and plasma cells produced during the lifetime of GC reaction confer long lasting protection [18], even after the termination of GC responses. Apart from the importance of GCs in natural infections, GCs are also the main target to promote antibody responses in vaccination. Therefore, strategies to enhance the GC responses and their efficiency are considered crucial for improving vaccination responses [19]. Adjuvants have been shown to alter the kinetics of GC reactions [20], suggesting the importance of the choice of adjuvant in vaccine development. Garg et al. developed an in silico GC model to investigate how the passively administered antibodies enhance humoral responses and predicted that the affinity of administered antibodies and antigen availability can be tuned therapeutically to optimize GC responses [21]. Extended antigen availability by slow delivery immunization also enhances GC and antibody responses [22,23,24] and promotes the development of broadly neutralizing antibodies [22]. Selection of B cells is also considered a target to induce the development of broadly neutralizing antibodies from GCs [25,26].

The GC lifetime is likely a critical factor in determining the extent of affinity maturation of B cells and efficiency of humoral immune response. Hence, it is important to gain an understanding of factors that determine the lifetime of GCs in order to identify potential therapeutic targets. Monitoring the kinetics of GC reactions throughout the overall lifetime of GCs has gained a lot of attention in the past [13,15,27,28] as changes in kinetics impact GC functionality [29,30]. Despite a large focus on the overall kinetics, lifetime of individual GCs composing the overall reaction and their dynamics are not studied, due to technical challenges in monitoring them for long time periods. In primary immune response, the time of appearance of individual GCs is variable resulting in an asynchronous onset of GCs [31,32], which could be synchronized to some extent by priming with a carrier protein prior to immunization. Data on the time course of GCs in spleen after SRBC immunization shows that new GCs are formed at least until 12 days after immunization [31]. Formation of new follicles with GCs in popliteal LNs were observed between day 5 to day 11 in the case of Keyhole limpet hemocyanin (KLH) immunization and between day 7 to day 14 in the case of phytohemagglutinin (PHA) immunization [33]. Some characteristic differences are also observed among the individual GCs such as the size which was found to be largely variable [14]. Individual GCs in newly formed follicles showed differences in the capacity of immune complex (IC) trapping and follicles with no or weak IC trapping capacity were also observed [33]. Furthermore, immunization with two different hapten carrier complexes resulted in differences in the antigen specificity of cells in the individual GCs, where 5–25% of the GCs had cells specific to one of the hapten only [15]. Clonal diversity and rates of homogenizing selection are also widely different among the individual GCs within the same LN [3,34]. However, the longevity and timeline of individual GCs composing the GC response are unknown.

Understanding the factors influencing lifetime of individual GCs in a primary immune response would also have implications in extending our knowledge of the shutdown of individual GCs and the overall GC response with implications for pathological GC deregulations. Considering the technical issues and complex behavior of GCs, mathematical modeling could provide great exploratory opportunity for testing different hypothesis and guide future experiments [35,36]. Using an agent-based model of the GC reaction, we analyzed the contraction phase of the primary immune response from experimental data to predict the timeline of individual GC reactions and identify various possibilities that are consistent with experimental data.

## 2. Materials and Methods

We investigated the longevity of individual GC reactions, starting with a hypothesis that asynchronously initialized GCs have similar lifetimes. A model was developed to simulate multiple GCs following an agent-based approach, where the individual cells of each GC are considered as agents. GC initiation times and other model parameters were chosen to reproduce the experimental GC dynamics. The hypothesis was further refined to consider GCs with variable lifetimes. Parameters and mechanisms leading to variability in GC lifetimes that are consistent with experimental GC dynamics were identified. In the proposed hypotheses, lifetime and plasma cell output of individual GCs were examined.

### 2.1. Model for Simulation of Multiple GCs

A computational framework was implemented that allows to simulate multiple asynchronous GCs (Figure 1). Individual GCs follow the basic characteristics of the agent-based GC simulation described in [37,38,39]. The base GC model was designed and parameterized to reproduce various experimental observations [37,39,40] such as spatial organization, B cell selection, affinity maturation, and clonal evolution [3,7]. The basic model included a 3D representation of GC by a discrete 3D lattice where the B cells proliferated, mutated, migrated, and were selected by interactions with follicular dendritic cells (FDCs) and T follicular helper cells (Tfh) cells in a spatio-temporal manner. GC space was equally divided into two compartments—light zone (LZ) and dark zone (DZ). CXCL12 and CXCL13 chemokine distributions were also established in the lattice. FDCs were placed in the LZ region and loaded with a fixed initial amount of 3000 antigen portions. Tfh cells were randomly incorporated in the lattice grids to initialize the model. During the simulation, Tfh cells responded to CXCL13 and tended to accumulate in the LZ. Seeder B cells were assumed to enter the GC space at a rate of 2 cells per hour for 96 h. The rate of influx was estimated [40] to reproduce the number of founder cells in [3]. Each seeder GC B cell divided a fixed number of times and then acquired an LZ phenotype. As the number of divisions of GC founder cells was unknown, six divisions were chosen which also corresponded to the number of divisions seen in response to stimulation of GC B cells with anti-DEC205-OVA [7]. Affinity of B cells was represented using a 4D shape space [41], where the position of a B cell with respect to a predefined optimal position was considered as a measure of BCR affinity. Somatic hypermutation was assumed to take place with a probability 0.5 [6,42] and lead to a random shift in the shape space position of the B cells. Unselected LZ B cells went through an antigen collection phase for 0.7 h, during which interaction with FDCs could result in antigen acquisition with a probability based on the BCR affinity. B cells that failed to collect antigen within this period underwent apoptosis. Upon successful antigen acquisition, LZ B cells interacted with the neighboring Tfh cells for a duration of 36 min. Multiple B cells were allowed to interact with the same Tfh cell at any timepoint and interacting Tfh polarized to the B cell that had collected the highest amount of antigen. At the end of the interaction time, a B cell was selected if Tfh was polarized to this B cell for at least 30 min, otherwise the B cell underwent apoptosis. Selection parameters were tuned to reproduce a mean number of divisions of GC B cells [9] and DZ–LZ ratio of GCs [7,37]. Selected cells underwent further rounds of division such that the number of divisions were determined by the amount of antigen collected by the B cell. Seventy two percent of the divisions were assumed to be asymmetric ([43] and Appendix A in [37]), wherein one daughter cell retained the collected antigen. Antigen-retaining B cells differentiated into plasma cells, and the remaining cells entered the unselected state to continue antigen collection. Detailed description of parameter values followed the supplementary text of [40]. Parameter changes with respect to this reference set are described in Appendix A. These parameter values were chosen to reproduce the experimental data mentioned in Section 2.2. We use the term Affinity Maturation (AM) response to refer to the collective GC response that constitutes all the GCs in a lymphoid organ to distinguish it from single GC reactions.

### 2.2. Experimental Data Used

In order to identify the timeline of GCs formed upon primary immunization, experimental data on dynamics of number of GCs from [31,44] were used. Multiple individual GCs were simulated corresponding to a time period of 40 days post immunization and the resulting change in number of GCs over time was compared to data. GC volume data from [13,45], were also used for comparison with simulation results.

### 2.3. Initiation of Representative GCs

Appearance of new GCs over time was assumed to follow a Hill function of the form
(1)$$N\left(t\right)=\frac{N_{\max}\;t^{n}}{T^{n}+t^{n}}$$
where N(t) is the cumulative number of GCs formed until time t, $N_{\max}$ is the maximum number of GCs formed, *T* is the time in hours when $N\left(t\right)=\frac{N_{\max}}{2}$ and n is the Hill coefficient.

The parameters of the Hill function $N_{\max}$, *T* and n were chosen such that the Hill function is consistent with the initial phase of the experimental data used (Appendix A). As the simulation of hundreds of GCs is computationally expensive, only a few representative GCs were simulated. Hence, the time period of appearance of new GCs is divided into many time intervals and 1 GC is simulated per time interval. The simulated GC is assumed to represent every GC initialized within the same time interval. The number of time intervals was chosen arbitrarily (15 in the case of extended formation of new GCs and 8 otherwise).

To allow some variation in the timing of initiation, the initiation time of simulated GCs was sampled from a Gaussian distribution with the mean equal to the midpoint of time interval described above and a width of 24 h. As simulated GCs in turn represent many other GCs initiated at similar time points, the other GCs were assumed to behave identically to the simulated one. This allowed us to study the effect of time shift caused by asynchronous onset without simulating hundreds of GCs.

### 2.4. Calculation of the Number of GCs and GC Lifetime

To estimate the number of GCs in simulation, a threshold for the number of B cells in a GC was arbitrarily set to 100 cells. Only GCs with a number of B cells above this threshold were counted. The lifetime of GC was considered as the time period during which the number of GC B cells remain above this threshold.

### 2.5. GC Simulations with Varying Antigen Availability

Exponential distributions were used to model the decrease in free ICs over time after immunization:(2)$$A\left(t\right)=A_{0}\;e^{-kt}$$
where A(t) is the number of antigen portions per FDC for GC initialized at time t and $A_{0}$ is the initial number of antigen portions at the time of immunization.

FDCs in the simulated GCs were loaded with an antigen concentration calculated from the exponential distribution at t = time of initiation. Parameter values other than the total antigen concentration on FDCs were the same for all simulated GCs. Parameter values in the exponential distribution used to fit the data were $A_{0}$ = 20,000, k = 0.026 for Rao data [31] and $A_{0}$ = 2000, k = 0.01 for Al-Qahtani data [44]. As the initiation time of GCs were sampled from Gaussian distributions, the antigen concentration acquired by these GCs also vary between simulations.

### 2.6. GC Simulations with Multiple Epitopes

Representation of multiple epitopes in the GC simulations follow the description in [46]. In the 4D shape space used for affinity representation, multiple epitopes were represented by multiple optimal positions at predefined points in the shape space. The fraction of different epitopes considered was reflected in the proportion of epitope availability among the total antigen concentration. Founder cell specificity was varied by choosing different affinities of founder cells with respect to the different epitopes or optimal positions in the shape space. In this study, the optimal positions were chosen far away which would correspond to epitopes that were unrelated [46].

### 2.7. Time Independent Random Variations

In the simulations with time independent random variations in parameters, the parameter value was assumed to vary between the simulated GCs. The parameter value for a GC was chosen from a Gaussian distribution, the mean and width of which was tuned in order to fit the data.

Simulations were performed using C++ [47] and each simulation was repeated 50–60 times. R [48] and ggplot2 package [49] were used for visualization of the simulation results. Cost of the simulation results ($S_{i}$) with respect to the data ($E_{i}$) was calculated as follows
(3)$$cost=\overset{n}{\underset{i=1}{\sum }}{\left(\frac{E_{i}-S_{i}}{E_{i}}\right)}^{2}$$

## 3. Results

### 3.1. Extended Formation of New GCs with Similar Lifetimes (Hypothesis 2) Is Consistent with Data

We developed an agent-based model of GC reaction (described in methods) able to simulate multiple co-evolving GCs and estimate the lifetime of individual GCs formed upon primary immunization. As GCs form asynchronously after a primary immunization, we considered such asynchronous GCs in the simulations corresponding to data from Rao et al. [31]. Few explicitly simulated GCs were assumed to represent multiple GCs. It is unknown how long new GCs are formed; however, the rise in the number of GCs in the data suggests that new GCs are formed at least for 12 days after immunization. The increase in the number of GCs was assumed to follow a Hill function and initiation time of simulated GCs were chosen as explained in methods such that the simulation result was consistent with the initial phase of data, during which an increase in number of GCs is observed.

Firstly, we assumed that the simulated GCs have similar characteristics (with small variability) and the parameters of the Hill function were chosen such that most GCs are formed within 12 days after immunization (Appendix A). This included simulating eight representative GCs (each representing 1–30 GCs for a section of spleen) initialized at different time points within 1–12 days after immunization (Appendix A). In this case, simulated GCs had similar lifetimes of approximately 12.5 days on average due to the similar characteristics (Appendix A). The regression phase showing a decline in the number of GCs in the simulation could not be fitted to the data with the above-mentioned assumptions (Appendix A). Instead, the number of GCs in the simulation decreased much more rapidly within a narrow time period compared to the data. On the other hand, a lifetime longer than 12.5 days could not explain the decrease in number of GCs seen in the data around day 12. This inability to fit the data suggested two possibilities: (1) New GCs might be formed for an extended period of time beyond the peak of the data, and/or (2) GCs might have highly variable lifetimes.

To test the first possibility, we extended the time period of formation of new GCs with similar characteristics by adaptation of the parameters of the Hill function for GC initiation (Appendix A). This possibility was able to fairly fit the experimental number of GCs [31] and the GC volume (data from [13,45]; Figure 2A,B). However, in order to be consistent with the data, a large number of GCs need to be initialized at late time points (until day 30–40 after immunization) (Appendix A and Figure 2C). According to this hypothesis, new GCs are continuously formed for an extended period of time and the contraction phase of AM response is prolonged due to the extended formation of new GCs. Due to the similar lifetime of individual GCs (approximately 10.5 days), the appearance of new GCs and their termination follow similar dynamics (Figure 2C,D). To the best of our knowledge, there is no convincing experimental evidence that new GCs are formed until the very end of the AM response. Therefore, we also tested the second possibility, where GCs formed have variable lifetimes.

### 3.2. Individual GCs Might Have Highly Variable Lifetimes

We hypothesized that the individual GCs might have variable lifetimes due to differences in certain unknown characteristics. As it is not understood what factors can lead to changes in the lifetime of GCs, we varied different parameters in the model to identify potential parameter changes that could lead to variability in the lifetime of GCs. We found that several parameters influencing B cell selection, such as the antigen availability, selection stringency by Tfh cells, recycling probability, and antibody feedback strength, were able to generate diverse GC lifetimes (Figure 3). GC lifetime varied weakly with changes in founder cell affinity (Figure 3E) and mutation probability (Figure 3F).

Although we cannot rule out the possibility that GCs have variable lifetimes due to a combination of these reasons or due to other unknown mechanisms, we tested if any of these characteristics could individually capture the extent of variability seen in the Rao data in order to fit the regression phase in the dynamics of the number of GCs. Among the hypothesis tested, three hypotheses were fairly consistent with the experimental data (summarized in Table 1).

### 3.3. Variability Due to Different Antigen Availability (Hypothesis 3)

The observation that free ICs are rapidly cleared from different organs after immunization [50,51] led to the hypothesis that individual GCs acquire different concentrations of antigens depending on the time of initiation. We assumed that the decrease in antigen as the GC formation was delayed followed an exponential function (Figure 4A). Such a time-dependent decrease in antigen trapping was able to generate variability in the lifetime of GCs and was consistent with the data (Figure 4B,C).

This hypothesis suggested that there could be GCs that lasted for the whole duration of the AM response, and there were GCs that prematurely terminated in a primary immune response. Early GCs survived longer and might extend throughout the lifetime of AM response. As the antigen concentration of late GCs was lower, these GCs shut down quickly and were short-lived compared to the early formed GCs (Figure 4E). This resulted in a typical distribution of GC lifetimes where there was a decrease in lifetime as the GCs were formed late (Figure 4D). The major producers of plasma cells were early formed GCs. Late GCs were compromised in their output production and did not contribute efficiently to the humoral response (Figure 4F). The prolonged contraction phase was due to the delayed shutdown of early formed GCs as opposed to the continuous formation of new GCs as discussed in the previous section. This hypothesis might be tested by measuring the antigen concentration in individual GCs. However, the challenge was in determining the age of GCs in such an analysis.

### 3.4. Lifetime Variability Due to Different Founder Cell Characteristics (Hypothesis 4)

As SRBC is a complex particle with multiple antigens and epitopes, we considered multiple epitopes in the simulation. Further, it has been shown that immunization with two different haptens could result in the formation of GCs where a significant proportion of GCs are specific to a single hapten only [15]. We hypothesized that the differences in lifetime seen in the data could be due to the presence of multiple epitopes and differences in the founder cell specificities of individual GCs towards the different epitopes. These assumptions were consistent with data when three unrelated epitopes were considered in different proportions. As the epitopes varied in abundance, the epitope with high concentration at the start of the simulation is referred as the dominant epitope.

This hypothesis was able to fit the data when early initialized GCs had random founders and late formed GCs had high affinity founders specific to a single epitope (Figure 5A,B). An alternative scenario, in which early formed GCs were initiated with specific founders and late formed GCs with random founder specificities, was also consistent with the data (Appendix A). Due to a lack of a deeper understanding of the mechanistic reason behind this hypothesis, the choice of founder cell specificities for the individual GCs could not be justified. Behavior of individual GCs in this case varied depending on the chosen founder cell specificity of individual GCs (Figure 5C–E). GCs having randomly chosen founder cells or highly specific founder cells towards the dominant epitopes were long-lived. GCs with founder cell specificity restricted to the least dominant epitopes were relatively short-lived (Figure 5C,D). Hence, the variability seen in the lifetime of GCs is due to the unequal availability of different epitopes and differences in the founder cell specificities. However, considering a single antigen epitope and founder cells with different affinities in the individual GCs was not consistent with the data, suggesting that changes in founder cell affinity are insufficient to explain the data (Table 1).

### 3.5. Initiation Time Independent Random Variation in GC Characteristics (Hypothesis 5)

In this hypothesis, we assumed that GCs have large random variation in the initial antigen amount irrespective of the initiation time of GCs. With GCs acquiring different antigen concentrations randomly sampled from a Gaussian distribution, the simulation results were fairly consistent with the experimental data (Figure 6A,B). In each simulation, GCs exhibited large variation in lifetime with respect to each other (Figure 6D) and there was no correlation between the initiation time and lifetime of GCs unlike the other hypotheses. However, with multiple simulation repeats, lifetimes of individual GCs were similar on average (Figure 6C), as the antigen concentration of GCs have large variation between simulations. This implies that such random changes in parameters could also be a reason for the prolonged contraction phase in the data.

### 3.6. Individual GCs Formed after NP-CGG Immunization

Previously described analysis was restricted to the data of Rao et al. [31]. However, GC reactions towards different antigen stimuli differ in their overall kinetics [16,54]. Hence, we performed a similar analysis on data obtained from spleen after NP-CGG immunization [44], which was qualitatively and quantitatively different from the data of SRBC immunization in [31]. The contraction phase of GCs was shorter when compared to data in [31], and the peak of the number of GCs appeared later. We found that assuming a heterogeneous lifetime among individual GCs was not necessary as the assumption of GCs with similar lifetime of approximately 17.5 days was able to fit the data (Appendix A). However, this data was also consistent with the other hypothesis tested with Rao data, although the lifetime of individual GCs and the extent of variability were different from those obtained by fitting Rao data (Figure 7). Although, we do not have concrete evidence that these GCs do have similar lifetimes, the lifetime of individual GCs and their variability might differ depending on immunization conditions as well, giving rise to characteristic changes in the overall kinetics.

## 4. Discussion

In this study, we predicted the longevity of individual GCs in spleen formed after primary immunization based on the existing knowledge of GC reactions. We showed that an asynchronous onset of GCs with similar lifetimes was unable to fit the data, if formation of new GCs was restricted to the initial phase of GC response. The prolonged contraction phase in the data suggested that new GCs might be formed until very late time points or that the GCs could have highly variable lifetimes. In previous experimental studies, persistence of cells with founder characteristics was observed throughout the AM response and is presumed to be due to longer influx of B cells into the GCs or the characteristics of cells which might have remained unchanged [55,56], rather than the continued formation of new GCs. While there is no experimental evidence suggesting that new GCs were continuously formed to replace the terminating GCs until a very late phase of the AM response, future studies are necessary to analyze how long new GCs are formed after immunization and the factors determining the initiation time of GCs. Although we provided an estimate for the initiation times of GCs, a detailed analysis of the mechanisms involved in GC formation and the initiation dynamics of GCs in a primary immune response might also reveal unknown mechanisms contributing to the variability among individual GCs.

Simulation of GCs with variable lifetimes predict that individual GCs might be long-lived which could exist for the entire duration of the AM response in a lymphoid organ or relatively short-lived that terminate within a few days. We found that considering differences in IC trapping and founder cell characteristics among individual GCs was consistent with the experimental data. Similar cost values with respect to the data suggests that these hypotheses cannot be distinguished. Decreased IC amount with increasing delay in initiation of GCs resulted in individual GCs with different lifetimes. Although primary follicles were capable of trapping ICs even before the GCs were formed [57], FDCs in the primary and secondary follicles differed morphologically [4] and the maturation state of FDCs might have affected the IC trapping ability [57]. In experiments where new follicles were induced to form after immunization, it was shown that IC trapping capacity of follicles might vary and the newly formed follicles with GCs had weak IC staining at the time of appearance of GCs [33]. Simulations with decreased antigen trapping predicted that late formed GCs might undergo shutdown quickly due to lack of antigen. Similarly, it was also observed that GCs that terminated early had a weak or no IC trapping capacity [33]. However, it is unknown whether the late formed GCs after SRBC immunization in the spleen were also associated with induction of new follicle formation. This hypothesis needs to be verified experimentally by measuring the ICs in individual GCs at different time points after immunization. Antigen amount has been shown to be correlated with the magnitude of Tfh and GC B cell responses [58], predicted to influence the affinity maturation [59] and tune the trade-off between quality and quantity of antibody responses [21]. Consistently, GCs with different IC amounts also showed differences in output in the simulations.

Our simulations suggest that differences in founder cell specificities towards unequally represented epitopes is another possibility that is able to explain the data. Although differences in antigen specificity of GC B cells were observed in individual GCs in the experiments, the reason for such differences is unknown. As GCs are formed over a period of time, it is reasonable to think that these GCs might have been founded with different kinds of clones either randomly or due to differences in the sequence of activation. It is also likely that late GCs were founded by memory cells produced from early GCs, and due to higher affinity of memory cells, founder cells of late GCs might have high affinities towards specific epitopes. Such differences in the founder cell composition might lead to differences in the characteristics and lifetime of late GCs compared to early formed GCs. Mesin et al., reported that the participation of memory cells in recall GCs is limited [60] and this can be explained by antibody feedback [46]. Future studies are necessary to address whether lower antibody feedback strength in a primary immune response, compared to recall responses, allows for the participation of memory cells in late stages of primary AM response. Moreover, IgM+ memory cells have been shown to participate in GC responses [61].

A large number of factors such as founder cell characteristics, antigen availability, selection stringency by Tfh cells, recycling probability, and strength of antibody feedback impacted the lifetime of GCs in silico. In addition to the hypothesis tested in this study, there could also be changes in the other parameters predicted either in an initiation time dependent or independent manner due to unknown mechanisms. In this context, our results also suggest that such time-independent random changes in parameters are a possibility. A more complex explanation including changes in several parameters among GCs cannot be excluded. Intercommunication between GCs is a possible mechanism which could lead to a more complex behavior of GCs. Passively administered soluble antibodies are able to regulate magnitude and kinetics of GCs, suggesting that endogenous antibodies from a GC might be able to enter other GCs in the same organism and affect the kinetics and output of individual GCs [52,53]. However, our analysis considering antibody feedback with single or multiple epitopes failed to fit the data when other differences between GCs are not considered (not shown) which suggests that factors other than antibody feedback strength might also be variable among the GCs. Despite this, antibody feedback might play an additional role in influencing the behavior of individual GCs, contributing to the variability in lifetimes, and this needs to be explored in the future. Recent findings suggest that Tfh cells are able to move between the GCs [62]. These observations suggest that GCs are not separate isolated entities and they might also be able to influence each other. Addressing the implications of information exchange between GCs could help researchers understand if the lifetime of a GC can also be modulated by the interaction with other GCs.

Dynamic changes are observed in GCs over a period of time [4]. Memory cells are predominantly produced from the GCs at early time points after immunization, but at later time points there is a switch towards largely producing plasma cells [63]. Considering multiple asynchronous GCs, it is unknown whether all GCs undergo such a time shift and contribute to both memory and plasma cells, or if GCs either producing only memory cells versus only plasma cells also exist. Late GC B cells might differ in characteristics and are not antigen dependent as depletion of FDCs did not have a large impact on GCs [64]. Similarly, it was suggested that high affinity B cells could be recruited after the GC is established [65]. The implications of such changes at the level of individual GCs are presently unknown.

Kinetics of AM responses have been found to vary depending on the antigen used, route and dynamics of administration, and the lymphoid organ under consideration [24,54,66,67]. IC trapping of individual GCs might vary depending on the nature of the antigen used for immunization and the kinetics of IC deposition on FDCs. The complexity of antigens such as the number of epitopes and associated founder cell composition also varies depending on the immunization conditions. This suggests that the lifetime of individual GCs and their variability might differ depending on the immunization agent used, and might be a contributing factor to the differences in the kinetics and lifetime of AM response observed experimentally under different conditions. It might be expected that AM response with an extended contraction phase or shorter contraction phase differ in the pattern of formation of new GCs and/or the lifetime of individual GCs. We validated the model extension with multiple GCs by comparing the simulation results to the data on overall volume kinetics of GCs and number of GCs. However, as there are considerable differences in GC kinetics under different immunization conditions, a systematic study on the evolution of number of GCs, overall GC volume and associated affinity maturation dynamics might help overcome the limited data availability and promote similar analysis for GC responses under a wide range of experimental conditions.

A better understanding of the lifetime of individual GCs is an important requirement to gain a deeper understanding of the termination of AM response. Monitoring individual GCs within the same organ for long periods of time can reveal a great deal of information in improving our understanding of individual GCs and facilitate the investigation of GC termination.

## Figures and Tables

**Figure 1 cells-10-01736-f001:**
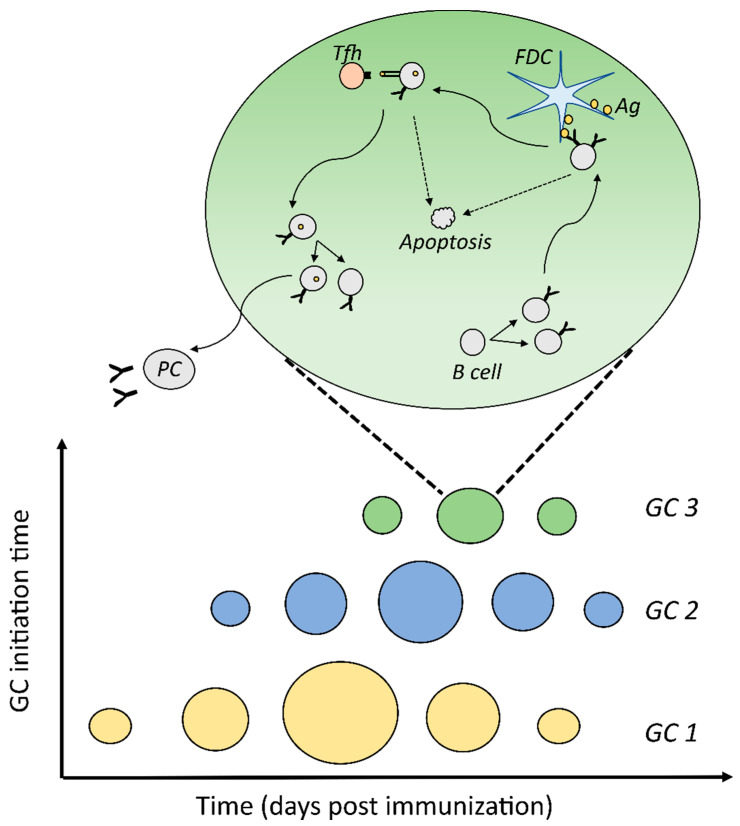
Schematic representation of the simulation of multiple asynchronous GCs. Individual GCs (*GC 1*, *GC 2*, and *GC 3*), were initialized at specific time points after immunization and were simulated for several days. Events taking place in individual GCs were modelled using an agent-based representation, where individual cells were simulated. In each GC, B cells proliferated, mutated their antibody genes, collected antigen from FDCs, interacted with Tfh cells, and underwent asymmetric divisions leading to differentiation of plasma cells. Lack of positive selection by FDCs and Tfh cells lead to B cell apoptosis. *GC*: germinal center; *FDC*: follicular dendritic cell; *Ag*: antigen; *PC*: plasma cell; Tfh: T follicular helper cell.

**Figure 2 cells-10-01736-f002:**
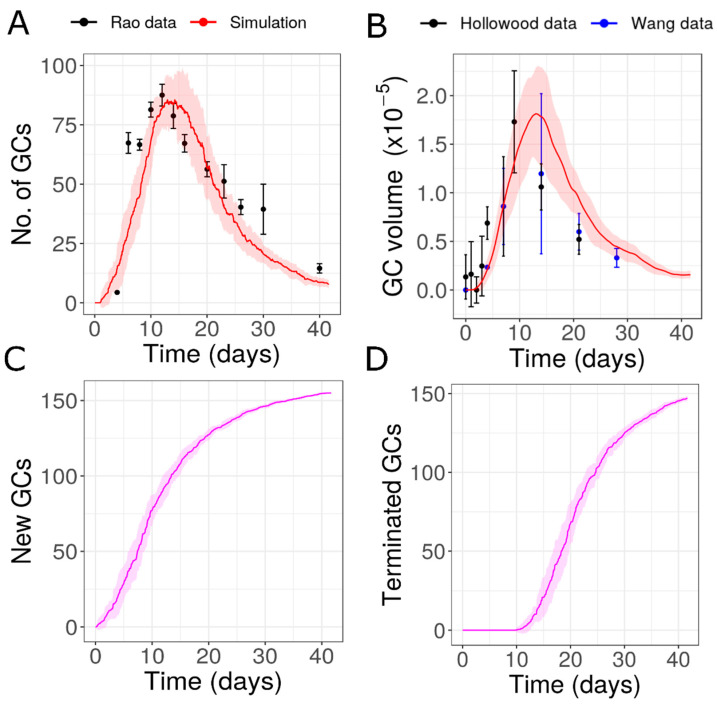
Simulation of GCs with similar characteristics formed over an extended period of time as shown in Appendix A: (**A**) Dynamics of number of GCs in simulation compared to data from [31], (**B**) total GC volume compared to data from [13,45] (**C**) cumulative number of new GCs initialized, (**D**) cumulative number of terminated GCs. GC volume data from [13,45] were normalized with respect to the simulation result on day 7. GC: Germinal Center.

**Figure 3 cells-10-01736-f003:**
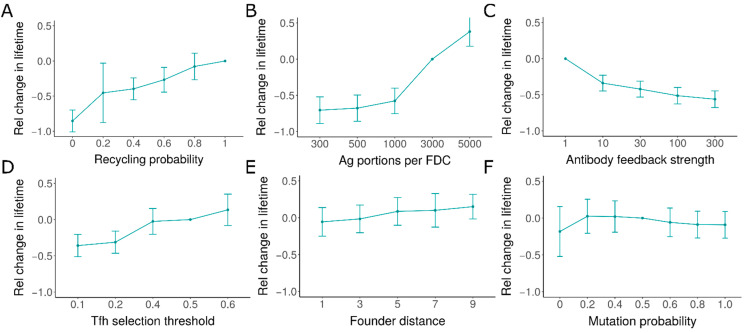
Variability in GC lifetime with varying parameter values: (**A**) probability of recycling, (**B**) antigen portions loaded in each FDCs, (**C**) strength of antibody feedback, (**D**) signal threshold required for selection by Tfh, (**E**) distance of founder cells in shape space from optimal position representing affinity to antigen, (**F**) mutation probability. Changes in lifetime were normalized with respect to the GC lifetime in simulation with reference parameter values (Appendix A). GC: germinal center, FDC: follicular dendritic cells, Tfh: T follicular helper cells.

**Figure 4 cells-10-01736-f004:**
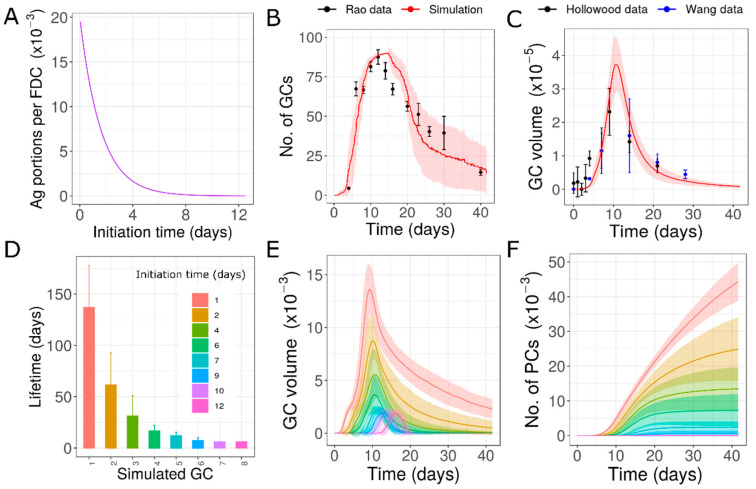
Simulations of GCs with antigen concentration depending on the time of appearance of GCs: (**A**) exponential function used to determine the antigen amount loaded in GCs as a function of starting time, (**B**) dynamics of number of GCs, (**C**) dynamics of total GC volume, (**D**) lifetime of simulated GCs, (**E**) GC volume of individual GCs simulated, (**F**) plasma cells produced from simulated individual GCs. In Panels D, E, and F, GCs initialized at different time points are shown in different colors. Experimental data from [13,31,45]. GC: germinal center; PC: plasma cell.

**Figure 5 cells-10-01736-f005:**
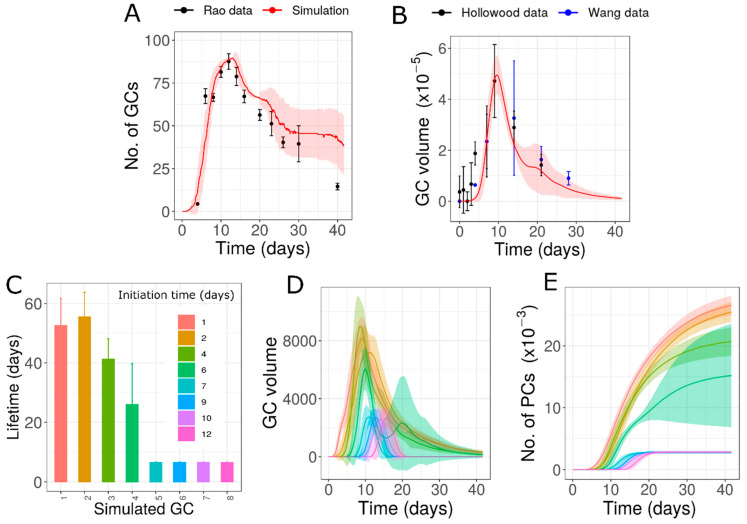
Simulations of GCs with multiple epitopes and different founder cell specificities: (**A**) dynamics of number of GCs, (**B**) dynamics of total GC volume, (**C**) lifetime of simulated GCs, (**D**) GC volume of individual GCs simulated, (**E**) plasma cells produced from simulated GCs. In Panels C, D and E, GCs initialized at different time points are shown in different colors. Experimental data from [13,31,45]. GC: germinal center; PC: plasma cell.

**Figure 6 cells-10-01736-f006:**
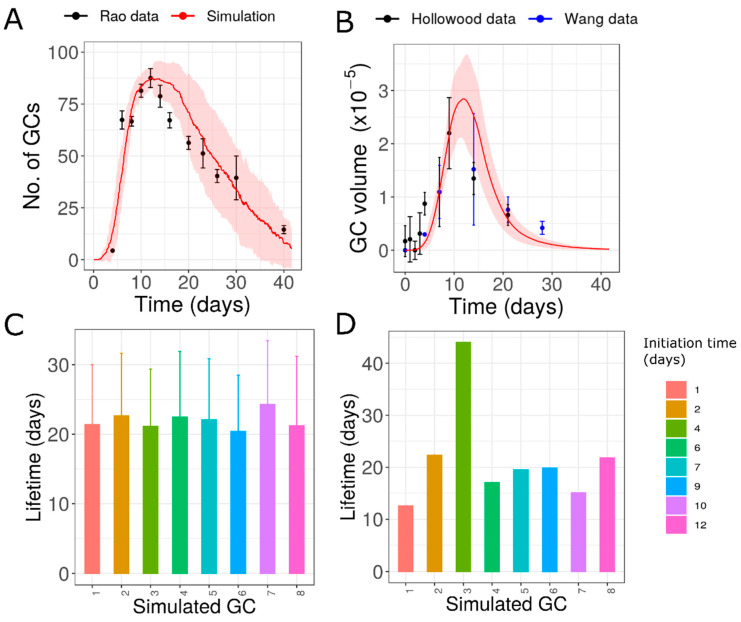
Simulations of GCs with different initial antigen concentrations independent of starting time: (**A**) dynamics of number of GCs, (**B**) total GC Volume, (**C**) lifetime of simulated GCs as an average of 60 simulations, (**D**) lifetime of simulated GCs in a single simulation. Experimental data from [13,31,45]. Color code in panels C and D, represent the initiation time of GCs. GC: Germinal Center.

**Figure 7 cells-10-01736-f007:**
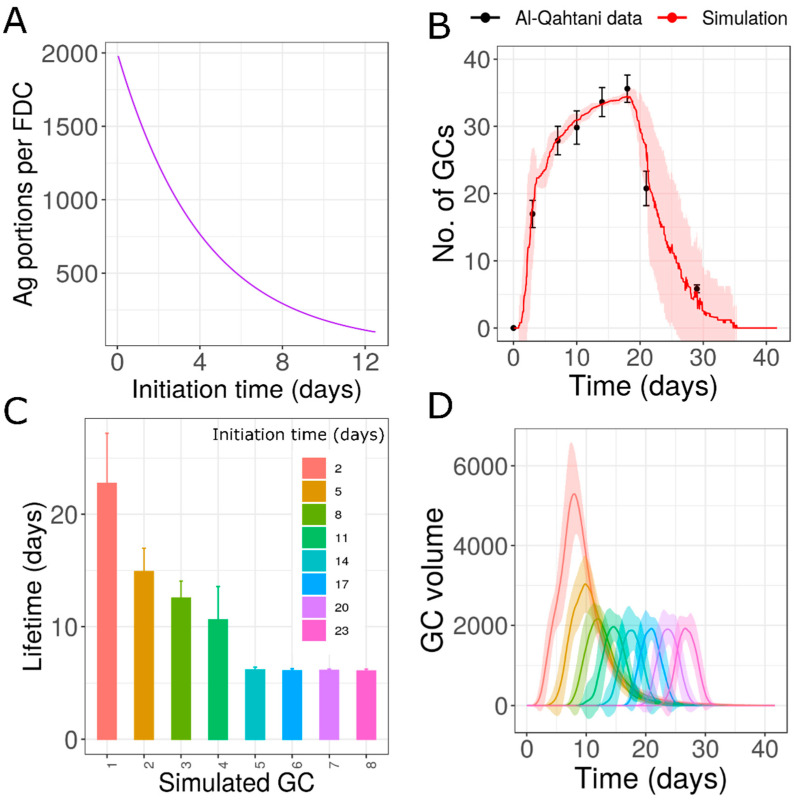
Simulation of GCs with different antigen concentrations as a function of initiation time as in Figure 4 but compared to data from Al-Qahtani et al. [44]. (**A**) Exponential function used to model the decrease in antigen concentration, (**B**) dynamics of number of GCs, (**C**) lifetime of simulated GCs, (**D**) GC volume of simulated GCs. Color code in panels C and D, represent the initiation time of GCs. GC: Germinal Center.

**Table 1 cells-10-01736-t001:** Summary of the hypothesis tested, motivation for the assumption and results obtained in fitting Rao data: Cost was calculated with respect to the experimental data on GC volume and number of GCs using Equation (3).

Hypothesis No.	Hypothesis	Motivation	Fitting the Data	Outcome	Cost
1	GCs with similar characteristics (with new GCs until 12 days post immunization)	Basic assumption	No	Simulated GCs shutdown within a narrow period of time	-
2	GCs with similar characteristics (with new GCs for extended time periods)	Unknown	Yes	Prolonged contraction phase due to extended formation of new GCs	10.777
3	Decrease in initial antigen concentration as GC formation is delayed	Rapid clearance of free antigen over time after immunization [50,51]	Yes	Early formed GCs are long-lived and late ones are short-lived	7.198
4	Multiple epitopes with different founder cell specificities	Due to memory cells seeding late formed GCs or differences in the sequence/timing of activation of different clones	Yes	Different lifetimes in GCs due to unequal epitope proportions and different specificities	11.645 or 7.262
5	Initiation time independent variation (antigen concentration)	Unknown	Yes	In each simulation, individual GCs have large variability in lifetimes when compared to each other. GC lifetime is not correlated with the initiation time	10.027
6	Antibody feedback with single or multiple epitopes	Antibodies might be exchanged between GCs and they can influence kinetics [52,53]	No	Does not exactly reproduce data despite resulting in some variability	-
7	No. of founder cells	Unknown	No	Insufficient lifetime variability	-
8	Founder cell affinities	Due to memory cells entering late formed GCs or differences in the sequence/timing of activation	No	Insufficient lifetime variability	-

## Data Availability

C++ code and R scripts used for this study are available at https://doi.org/10.5281/zenodo.5012498.

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
