# Peer review of "In Silico Analysis of the Longevity and Timeline of Individual Germinal Center Reactions in a Primary Immune Response"

_cells, 2021, doi:10.3390/cells10071736_

Round 1

Reviewer 1 Report

/

Author Response

We thank the reviewer for the time in reviewing this manuscript.

Reviewer 2 Report

The manuscript titled “In-silico analysis of the longevity and timeline of individual germinal centre reactions in a primary immune response” focuses on the study of germinal centres (GCs) and the lifetime of individual GCs. The authors argue how the in-silico approach could overcome all the technical difficulties present when several individual GCs are longitudinally monitored.

The scientific importance of such a study is certainly very high. It could improve the investigation of specific parameters of individual GCs in lymphoid organs to demonstrate the hypotheses that the in-silico investigation proved more probable.

However, I have some minor comments and concerns which may help to improve the manuscript:

Comment #1

In paragraph “2.1. Model for simulation of multiple GCs”, to better understand the hypotheses, a graphical schematic representation of the model would be necessary. The representation could be added as a figure in the article or added in supplementary figures. 

Comment #2

Table 1 represents all the parameter values used in the simulations of different hypotheses, but it isn't easy to read. I would suggest changing the orientation of the table to landscape orientation. In that way, also hypothesis no. 3 could be in just one line and better readable.

Comment #3

Page 7, line 245. The correct mention is Figure 2E instead of Figure 2EF.

Comment #4

Figure2. I would suggest changing the scale of the figure using the range -1.00 to 0.5. A similar scale in lifetime change could permit to compare better the factors presented by the authors and discriminate the important ones from the others.

Comment #5

Table 2. I would suggest adding the number to the hypotheses represented in the table to better follow the hypotheses explained in the next paragraphs. Moreover, there is a typo in the sentence “No of founder cells.” please correct it in “No. of founder cells”.

Comment #6

Figure 3. Here and in the next similar figures, it is necessary to represent the legend of colors and time points when GCs were initialized (Fig. 3E, Fig. 3F). Moreover, it would be nice to have the same colors also in the representation of figure 3D.

Comment #7

In the result part, it would be useful for readers to have directly in the title of the sub-paragraph the number of the hypothesis which is explained there.

Comment #8

I would suggest that the authors upload in the supplementary data both the C++ and R scripts that they used for the analyses. It could improve the sharing of the information with other scientists and refine the hypotheses suggested by the authors by all the scientists that could have a well-defined shape of the structure of the analysis.

Author Response

We thank the reviewer for the comments and suggestions.

  1. Thanks for the suggestion, we have added a graphical representation of the model as Figure 1 in the manuscript.
  2. We have changed the orientation of the table to landscape.
  3. Thanks for pointing to this error, we have corrected this.
  4. We have changed the y-axis ranges in the figure as suggested.
  5. We have enumerated each hypothesis and corrected the typo in the table.
  6. We have added a legend and also changed the colors of the bar plot representing GC lifetimes in all the figures.
  7. We have included the hypotheses number in the sub-titles of results section.
  8. We have uploaded the code and scripts used for this study in Zenodo.

https://doi.org/10.5281/zenodo.5012498

Reviewer 3 Report

Arulraj et al. presented a computational study using an agent-based modeling approach to investigate the lifetime of germinal centers in the primary immune response. The authors found the variation in germinal centers' lifetime that they predicted could arise because of changes in antigen availability and cell composition. The authors presented an exciting application of modeling. However, I have several concerns. Major: (A). "Using an agent-based model of the GC reaction, we analyzed the contraction phase of the primary immune response..." Why did the authors focus only on the contraction phase of the primary immune response? (B). During model building, the authors used many assumptions without providing any background and rationale. It is tough to understand why the authors chose these parameters and values. Please justify all the parameters assumed during model building and simulations. Also, add the relevant citations. As a sample, I am providing a few below from section 2 1. "FDCs are placed in the LZ region and loaded with a fixed initial concentration of antigen." What are these concentrations, and how did the authors determine them? 2. "Tfh cells are incorporated randomly in the lattice grids." What is the rationale behind the random incorporation of Tfh cells? 3. "Seeder B cells are assumed to enter the GC space at a rate of 2 cells per hour for 96 hours. Each seeder GC B cell divides six times and then acquires a LZ phenotype." Please provide background and rationale for these assumptions. In addition, provide citations if these numbers are obtained from some experimental studies. 4. "Unselected LZ B cells go through an antigen collection phase for 0.7 h.." Please provide rationale. 5. "LZ B cells interact with the neighboring Tfh cells for a duration of 36 minutes.." Please provide a rationale for assuming 36 minutes. 6. "At the end of interaction time, a B cell is selected if Tfh was polarized to this B cell for at least 30 minutes, otherwise the B cell undergoes apoptosis." How did the authors come up with 30 minutes? Please provide rationale. (C). How did the authors validate the model? Is the model sensitive/robust to changes in the parameters or initial concentration? Minor: Minor: (A). In material and methods, the authors directly presented the model. However, a few sentences about the whole modeling process would be helpful for readers. (B). There are too many abbreviations used in the article which have a full name at first appearance. It is tough to keep track of all of them. For example, the full name of FDCs was mentioned on page 1; the abbreviation reappears on Page 3. It is okay to use abbreviations, but the authors should write full names (with abbreviations) here and there that might help readers to remember them.

Author Response

We thank the reviewer for the comments and suggestions.

Major A) Timing of initiation and shutdown of individual GCs are both unknown. We focused on identifying the factors that influence the shutdown of GCs and thus impact GC lifetime rather than the factors governing the onset of GCs. So, we chose to fit the initiation times from the data and predicted the GC lifetimes, such that the contraction phase in the simulation is consistent with that of the data. However, analyzing the initiation dynamics of GCs and potential reasons behind the asynchronous onset of GCs can be considered as a future perspective of this study and we highlighted this in the discussion part of the manuscript by including the following sentence.

“Although, we provided an estimate for the initiation times of GCs, a detailed analysis of the mechanisms involved in GC formation and the initiation dynamics of GCs in a primary immune response might also reveal unknown mechanisms contributing to the variability among individual GCs”.

B) The basic elements of the agent-based model used in this study is previously published [Meyer-Hermann, Cell Reports 2012; Meyer-Hermann, JImmunol 2014; Binder and Meyer-Hermann, Front.Immunol 2016]. As several of the parameter values are not directly measurable in the experiments, this model was parameterized such that various properties including spatial organization, affinity maturation and clonal evolution are consistent with the data and show acceptable behavior. For instance, properties of FDCs and Tfh cells are chosen so as to mimic the spatial organization seen in a mature GC, where FDC network and Tfh cells are mostly present in the light zone. Although the Tfh cells are randomly introduced initially, they tend to accumulate in the LZ as they respond to CXCL13.

We have added the following sentence in the manuscript to better explain this.

“The base GC model was designed and parameterized to reproduce various experimental observations [Meyer-Hermann et al, Cell Reports 2012; Binder and Meyer-Hermann, Front Immunol 2016; Meyer-Hermann et al, Front Immunol 2018] such as spatial organization, B cell selection, affinity maturation and clonal evolution (Victora et al, Cell 2010, Tas et al, Science 2016).”

C) We have added the following sentences in the methods and discussion sections to address the points related to model validation.

“The base GC model was designed and parameterized to reproduce various experimental observations [Meyer-Hermann et al, Cell Reports 2012; Binder and Meyer-Hermann, Front Immunol 2016; Meyer-Hermann et al, Front Immunol 2018] such as spatial organization, B cell selection, affinity maturation and clonal evolution (Victora et al, Cell 2010, Tas et al, Science 2016).” 

“We validated the model extension with multiple GCs by comparing the simulation results to the data on overall volume kinetics of GCs and number of GCs. However, as there are considerable differences in GC kinetics under different immunization conditions, a systematic study on the evolution of number of GCs, overall GC volume and associated affinity maturation dynamics might help overcome the limited data availability and promote similar analysis for GC responses under a wide range of experimental conditions.”

Regarding the robustness/sensitivity of the results to the parameter values, we have tested the sensitivity of GC lifetimes to changes in parameter values and found that parameters such as antigen concentration and recycling probability have a relatively large impact on GC lifetime compared to parameters such as mutation probability. This is shown in Figure 3.

Minor A) We have added the following sentences at the beginning of the methods section to explain the modeling process.

“We investigated the longevity of individual GC reactions, starting with a hypothesis that asynchronously initialized GCs have similar lifetimes. A model was developed to simulate multiple GCs following an agent-based approach, where the individual cells of each GC are considered as agents. GC initiation times and other model parameters were chosen to reproduce the experimental GC dynamics. The hypothesis was further refined to consider GCs with variable lifetimes. Parameters and mechanisms leading to variability in GC lifetimes that are consistent with experimental GC dynamics were identified. In the proposed hypotheses, lifetime and plasma cell output of individual GCs were examined.”

B) As suggested, we have reintroduced some of the abbreviations.

Round 2

Reviewer 3 Report

However, the authors have addressed some of the comments from my previous review. They have overlooked Major concerns in point (B). 
Please revise the manuscript considering all the comments and provide a point-by-point response. I am appending the comments from my previous review below. 

(B). During model building, the authors used many assumptions without providing any background and rationale. It is tough to understand why the authors chose these parameters and values. Please justify all the parameters assumed during model building and simulations. Also, add the relevant citations. As a sample, I am providing a few below from section 2 1. "FDCs are placed in the LZ region and loaded with a fixed initial concentration of antigen." What are these concentrations, and how did the authors determine them? 
2. "Tfh cells are incorporated randomly in the lattice grids." What is the rationale behind the random incorporation of Tfh cells? 
3. "Seeder B cells are assumed to enter the GC space at a rate of 2 cells per hour for 96 hours. Each seeder GC B cell divides six times and then acquires a LZ phenotype." Please provide background and rationale for these assumptions. In addition, provide citations if these numbers are obtained from some experimental studies. 
4. "Unselected LZ B cells go through an antigen collection phase for 0.7 h.." Please provide rationale. 
5. "LZ B cells interact with the neighboring Tfh cells for a duration of 36 minutes.." Please provide a rationale for assuming 36 minutes. 
6. "At the end of interaction time, a B cell is selected if Tfh was polarized to this B cell for at least 30 minutes, otherwise the B cell undergoes apoptosis." How did the authors come up with 30 minutes? Please provide rationale.

Author Response

We would like to thank the reviewer again for suggestions in improving the model description and apologize for not satisfactorily addressing these comments previously. We have now elaborated the response and revised the manuscript accordingly.

  1. Antigen amount used in the base GC model was 3000 portions per FDC [reference parameter set in Table 1]. In this study, antigen concentration was re-parameterized under different hypothesis such that the GC volume kinetics in the simulation reproduces the data from [Rao et al, 2002; Al-Qahtani et al, 2008; Hollowood and Macartney, 1992; Wang and Carter 2005]. These concentrations (as well as the other parameters) are provided in Table 1.

We have rephrased the following sentence in the manuscript

“FDCs are placed in the LZ region and loaded with a fixed initial amount of 3000 antigen portions.”

We have also added the following sentence.

“Parameter changes with respect to this reference set are described in Table 1. These parameter values were chosen to reproduce the experimental data mentioned in section 2.2”.

  1. Random incorporation of Tfh cells is an arbitrary choice to initialize the model. Later in the simulation, known spatial distribution of Tfh cells in the light zone is achieved by chemotaxis. We have rephrased this sentence in the manuscript as follows.

“Tfh cells are randomly incorporated in the lattice grids to initialize the model. During the simulation, Tfh cells respond to CXC13 and tend to accumulate in the LZ.”

  1. We have added the following sentences in the manuscript to address this point.

“As the number of divisions of GC founder cells is unknown, six divisions were chosen which also corresponds to the number of divisions seen in response to stimulation of GC B cells with anti-DEC205-OVA [Victora 2010].”

“The rate of influx was estimated [Meyer-Hermann et al., Front Immunol 2018] to reproduce the number of founder cells in [Tas et al., Science 2016].”

4-6. As mentioned in the previous response, many of the parameters were chosen in the model to reproduce various experimental observations such as clonal diversity, spatial organization [Meyer-Hermann et al, Cell Reports 2012; Binder and Meyer-Hermann, Front Immunol 2016; Meyer-Hermann et al, Front Immunol 2018]. The parameters antigen collection period (point 4), Tfh-B cell interaction period (point 5) and minimum polarization period (point 6), were also chosen in this way. Therefore, we could not state a rationale for these parameters individually but added the following sentence in the manuscript.  

“Selection parameters were tuned to reproduce mean number of divisions of GC B cells [Gitlin et al 2014] and DZ-LZ ratio of GCs [Victora 2010, Meyer-Hermann 2012].”

However, we have provided relevant citations for parameter values that were obtained from experimental data.